# Action Inference by Maximising Evidence: Zero-Shot Imitation from Observation with World Models

**Xingyuan Zhang[1,2]\*, Philip Becker-Ehmck[1], Patrick van der Smagt[1], Maximilian Karl[1]**
[1]Machine Learning Research Lab, Volkswagen Group, [2]Technical University of Munich
`{xingyuan.zhang,philip.becker-ehmck,karlma}@argmax.ai`

## Abstract

Unlike most reinforcement learning agents which require an unrealistic amount of environment interactions to learn a new behaviour, humans excel at learning quickly by merely observing and imitating others. This ability highly depends on the fact that humans have a model of their own embodiment that allows them to infer the most likely actions that led to the observed behaviour. In this paper, we propose Action Inference by Maximising Evidence (AIME) to replicate this behaviour using world models. AIME consists of two distinct phases. In the first phase, the agent learns a world model from its past experience to understand its own body by maximising the evidence lower bound (ELBO). While in the second phase, the agent is given some observation-only demonstrations of an expert performing a novel task and tries to imitate the expert's behaviour. AIME achieves this by defining a policy as an inference model and maximising the evidence of the demonstration under the policy and world model. Our method is "zero-shot" in the sense that it does not require further interactions with the environment after given the demonstration. We empirically validate the zero-shot imitation performance of our method on the Walker of the DeepMind Control Suite and find it outperforms the state-of-the-art baselines. We also find AIME with image observations still matches the performance of the baseline observing the true low-dimensional state of the environment.

## 1 Introduction

In recent years, deep reinforcement learning (DRL) has enabled intelligent decision-making agents to thrive in multiple fields Mnih et al. (2015); Silver et al. (2016); Vinyals et al.; Choi et al. (2023); OpenAI et al. (2019); Ouyang et al. (2022). However, one of the biggest issues of DRL is sample inefficiency. The dominant framework in DRL is learning from scratch Agarwal et al. (2022). Thus, most algorithms require an incredible amount of interactions with the environment Mnih et al. (2015); Silver et al. (2016); Vinyals et al..

In contrast, cortical animals such as humans are able to quickly learn new tasks through just a few trial-and-error attempts, and can further accelerate their learning process by observing others. An important difference between biological learning and the DRL framework is that the former uses past experience for new tasks. When we try a novel task, we use previously learnt components, and generalise to solve the new problem efficiently. This process is augmented by imitation learning Iacoboni (2008), which allows us to replicate similar behaviours without direct observation of the underlying muscle movements. If the DRL agents could similarly harness observational data, such as the abundant online video data, the sample efficiency may be dramatically improved Baker et al. (2022).

However, directly learning a model from observation-only sequences is insufficient for both biological and technical systems, since we can only observe the outcome of actions, but do not know the actions themselves. Without knowing these actions, the observation sequences are highly stochastic and multimodal Babaeizadeh et al. (2018). Trying to infer these unknown actions without prior knowledge of

---

\*Corresponding author.

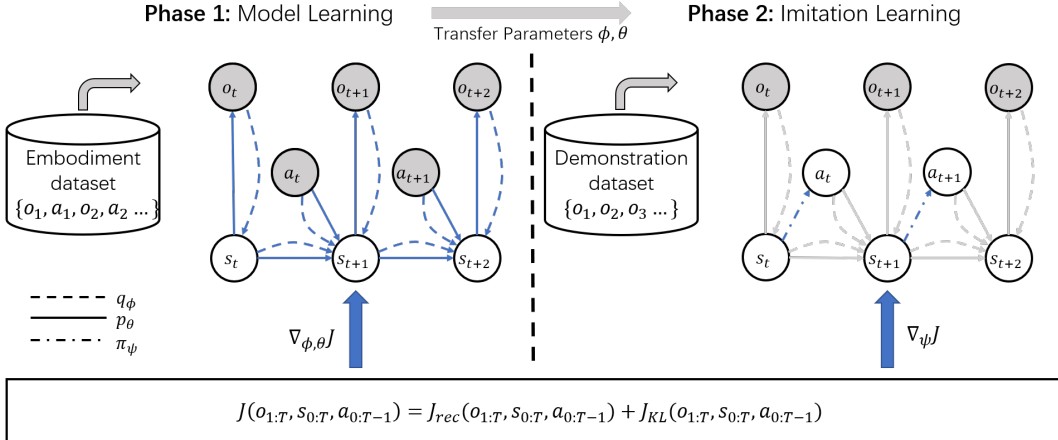

Figure 1: Overview of AIME algorithm. In phase 1, both observations and actions are provided by the embodiment dataset and the agent learns a variational world model to model the evidence of observations conditioned on the actions. Then the learnt model weights are frozen and transferred to phase 2. In phase 2, only the observations are provided by the demonstration dataset, so the agent needs to infer both states and actions. The action inference is achieved by the policy model which samples actions given a state. The grey lines indicate the world model parameters are frozen in phase 2. Both phases are optimised toward the same objective, i.e. the ELBO.

the world is difficult due to the problem of attributing which parts of the observations are influenced by the actions and which parts are governed by normal system evolution or noise.

Therefore, in this work, we hypothesise that in order to make best use of observation-only sequences, an agent has to first understand the notion of an action. This can be achieved by learning a model from an agent's past experiences where both the actions and their consequences, i.e. observations, are available. Given such a learnt model which includes a causal model of actions and their effects, it becomes feasible for an agent to infer an action sequence leading to given observation-only data.

In this work, we propose a novel algorithm, Action Inference by Maximising Evidence (AIME), to try to replicate the imitation ability of humans. The agent first learns a world model from its past experience by maximising the evidence of these experiences. After receiving some observation-only demonstrations of a novel task, the agent tries to *mimic* the demonstrator by finding an action sequence that makes the demonstration most likely under the learnt model. This procedure is shown in Figure 1.

Our contribution can be summarised as follows:

- We propose AIME, a novel method for imitation from observation. AIME first learns a world model by maximising the evidence of its past experience, then consider the policy as an action inference model and imitates by maximising the evidence of demonstration.
- We conduct experiments with a variety of datasets and tasks to demonstrate the superior performance of AIME compared with other state-of-the-art methods.

## 2 PROBLEM FORMULATION

Consider an MDP problem defined by the tuple $\{S, A, T, R\}$, where $S$ is the state space, $A$ is the action space, $T : S \times A \to S$ is the dynamic function and $R : S \to \mathbb{R}$ is the reward function. A POMDP adds partial observability upon an MDP with two components: the observation space $O$ and the emission function $\Omega : S \to O$. The six components of a POMDP can be categorised into three groups: $S$, $A$ and $T$ define the embodiment of our agent, $O$ and $\Omega$ define the sensors of our agent and $R$ itself defines the task. The goal is to find a policy $\pi : S \to A$ which maximises the accumulated reward, i.e. $\sum_t r_t$.

In this paper, we want to study imitation learning within a fixed embodiment across different tasks. We presume the existence of two datasets for the same embodiment:

- Embodiment dataset $D_{\text{body}}$ contains trajectories $\{o_0, a_0, o_1, a_1 \dots\}$ that represent past experiences of interacting with the environment. This dataset provides information about the embodiment for the algorithm to learn a model. For example, in this paper, the dataset is a replay buffer filled while solving some tasks with the same embodiment. But in general, it may be any collection of past experiences of the embodiment.

- Demonstration dataset $D_{\text{demo}}$ contains a few expert trajectories $\{o_0, o_1, o_2 \dots\}$ of the embodiment solving a certain task defined by $R_{\text{demo}}$. The crucial difference between this dataset and the embodiment dataset is that the actions are not provided anymore since they are not observable from a third-person perspective.

The goal of our agent is to use information in $D_{\text{body}}$ to learn a policy $\pi$ from $D_{\text{demo}}$ which can solve the task defined by $R_{\text{demo}}$ as well as by the expert who generated $D_{\text{demo}}$. For the sake of simplicity, we assume that the two datasets share the same observation space $O$ and the emission model $\Omega$.

## 3 METHODOLOGY

In this section, we describe our proposed method, AIME, in detail. AIME consists of two phases. In the first phase, the knowledge of the embodiment is learnt through a form of world model; while in the second phase, this knowledge is used to imitate the expert.

### 3.1 PHASE 1: MODEL LEARNING

In the first phase, we need to learn a model to understand our embodiment. We choose to achieve this by learning a world model. As an analogy to a language model, we formally define a world model as a probability distribution over sequences of observations. The model can be either unconditioned or conditioned on other factors such as previous observations or actions. For phase 1, the model needs to be the conditional distribution, i.e. $p(o_{1:T}|a_{0:T-1})$, to model the effect of the actions. When given a certain observation sequence, the likelihood of this sequence under the model is usually referred to as evidence.

In this paper, we consider variational world models where the observation is governed by a Markovian hidden state. In the literature, this type of model is also referred to as a state-space model (SSM). Such a variational world model involves four components, namely

$$
\begin{aligned}
\text{encoder } z_t &= f_\phi(o_t),\\
\text{posterior } s_t &\sim q_\phi(s_t|s_{t-1}, a_{t-1}, z_t),\\
\text{prior } s_t &\sim p_\theta(s_t|s_{t-1}, a_{t-1}),\\
\text{decoder } o_t &\sim p_\theta(o_t|s_t).
\end{aligned}
$$

$f_\phi(o_t)$ is the encoder to extract the features from the observation; $q_\phi(s_t|s_{t-1}, a_{t-1}, z_t)$ and $p_\theta(s_t|s_{t-1}, a_{t-1})$ are the posterior and the prior of the latent state variable; while $p_\theta(o_t|s_t)$ is the decoder that decodes the observation distribution from the state. $\phi$ and $\theta$ represent the parameters of the inference model and the generation model respectively.

Typically, a variational world model is trained by maximising the ELBO which is a lower bound of the log-likelihood, or evidence, of the observation sequence, i.e. $\log p_\theta(o_{1:T}|a_{0:T-1})$. Given a sequence of observations, actions, and states, the objective function can be computed as

$$J(o_{1:T}, s_{0:T}, a_{0:T-1}) = J_{\text{rec}}(o_{1:T}, s_{0:T}, a_{0:T-1}) + J_{\text{KL}}(o_{1:T}, s_{0:T}, a_{0:T-1}), \tag{1}$$

$$\text{where } J_{\text{rec}}(o_{1:T}, s_{0:T}, a_{0:T-1}) = \sum_{t=1}^{T} \log p_\theta(o_t|s_t), \tag{2}$$

$$J_{\text{KL}}(o_{1:T}, s_{0:T}, a_{0:T-1}) = \sum_{t=1}^{T} -D_{\text{KL}}[q_\phi(s_t|s_{t-1}, a_{t-1}, f_\phi(o_t))||p_\theta(s_t|s_{t-1}, a_{t-1})]. \tag{3}$$

The objective function is composed of two terms: the first term $J_{\text{rec}}$ is the likelihood of the observation under the inferred state, which is usually called the reconstruction loss; while the second term $J_{\text{KL}}$ is the KL divergence between the posterior and the prior distributions of the latent state. To compute the

objective function, we need to sample the inferred states from the observation and action sequence. This is done by autoregressively sampling from the posterior with the re-parameterisation trick Kingma & Welling (2014); Rezende et al. (2014).

Combining all these, we formally define the optimisation problem for this phase as

$$\phi^*, \theta^* = \operatorname*{argmax}_{\phi, \theta} \mathbb{E}_{\{o_{1:T}, a_{0:T-1}\} \sim D_{\text{body}}, s_{0:T} \sim q_\phi} [J(o_{1:T}, s_{0:T}, a_{0:T-1})]. \tag{4}$$

In this work, we use a specific variational world model called RSSM Hafner et al. (2019; 2020), which offers state-of-the-art performances by splitting the latent state to be a combination of deterministic and stochastic components.

## 3.2 PHASE 2: IMITATION LEARNING

In the second phase, we want to utilise the knowledge of the world model from the first phase to imitate the expert behaviour from the demonstration dataset $D_{\text{demo}}$ in which only sequences of observations but no actions are available. We will derive our algorithm from two different perspectives.

**The Bayesian derivation** Since the actions are unknown in the demonstration, instead of modelling the conditional evidence in phase 1, we need to model the unconditional evidence, i.e. $\log p_\theta(o_{1:T})$. Thus, we also need to model the actions as latent variables together with the states. In this way, the reconstruction term $J_{\text{rec}}$ will stay the same as eq. (2), while the KL term will be defined on the joint distribution of states and actions, i.e.

$$J_{\text{KL}}(o_{1:T}, s_{0:T}, a_{0:T-1}) = \sum_{t=1}^{T} -D_{\text{KL}}[q_{\phi,\psi}(s_t, a_{t-1}|s_{t-1}, f_\phi(o_t)) || p_{\theta,\psi}(s_t, a_{t-1}|s_{t-1})]. \tag{5}$$

If we choose the action inference model in the form of a policy, i.e. $\pi_\psi(a_t|s_t)$, and share it in both posterior and prior, then the new posterior and prior can be factorised as

$$q_{\phi,\psi}(s_t, a_{t-1}|s_{t-1}, f_\phi(o_t)) = \pi_\psi(a_{t-1}|s_{t-1})q_\phi(s_t|s_{t-1}, a_{t-1}, f_\phi(o_t)) \tag{6}$$

$$\text{and } p_{\theta,\psi}(s_t, a_{t-1}|s_{t-1}) = \pi_\psi(a_{t-1}|s_{t-1})p_\theta(s_t|s_{t-1}, a_{t-1}) \tag{7}$$

respectively. When we plug them into the eq. (5), the policy term cancels and we will get a similar optimisation problem with phase 1 as

$$\psi^* = \operatorname*{argmax}_{\psi} \mathbb{E}_{o_{1:T} \sim D_{\text{demo}}, \{s_{0:T}, a_{0:T-1}\} \sim q_{\phi^*, \psi}} [J(o_{1:T}, s_{0:T}, a_{0:T-1})]. \tag{8}$$

The main difference between eq. (4) and eq. (8) is where the action sequence is coming from. In phase 1, the action sequence is coming from the embodiment dataset, while in phase 2, it is sampled from the policy instead since it is not available in the demonstration dataset.

**The control derivation** From another perspective, we can view phase 2 as a control problem. One crucial observation is that, as shown in eq. (1), given a trained world model, we can evaluate the lower bound of the evidence of any observation sequence given an associated action sequence as the condition. In a deterministic environment where the inverse dynamics model is injective, the true action sequence that leads to the observation sequence is the most likely under the true model. In general, the true action sequence may not necessarily be the most likely under the model. This is, however, a potential benefit of our approach. We are mainly interested in mimicking the expert's demonstration and may be better able to do so with a different action sequence.

Thus, for each observation sequence that we get from the demonstration dataset, finding the missing action sequence can be considered as a trajectory-tracking problem and can be tackled by planning. To be specific, we can find the missing action sequence by solving the optimisation problem

$$a_{0:T-1}^* = \operatorname*{argmax}_{a_{0:T-1}} \mathbb{E}_{o_{1:T} \sim D_{\text{demo}}, s_{0:T} \sim q_{\phi^*}} [J(o_{1:T}, s_{0:T}, a_{0:T-1})]. \tag{9}$$

If we solve the above optimisation problem for every sequence in the demonstration dataset, the problem will be converted to a normal imitation learning problem and can be tackled with standard techniques such as behavioural cloning. We can also view this as forming an implicit inverse dynamics model (IDM) by inverting a forward model.

---

**Algorithm 1:** AIME

---

**Data:** Embodiment dataset $D_{\text{body}}$, Demonstration dataset $D_{\text{demo}}$, Learning rate $\alpha$

*# Phase 1: Model Learning*

Initialise world model parameters $\phi$ and $\theta$

**while** *model has not converged* **do**

    $\{o_{1:T}, a_{0:T-1}\} \sim D_{body}$

    $s_0 \leftarrow 0$

    **for** $t = 1 : T$ **do**

        $s_t \sim q_\phi(s_t | s_{t-1}, a_{t-1}, f_\phi(o_t))$

    Compute objective function $J$ from eq. (1)

    Update model parameters $\phi \leftarrow \phi + \alpha \nabla_\phi J, \theta \leftarrow \theta + \alpha \nabla_\theta J$

*# Phase 2: Imitation Learning*

Initialise policy parameters $\psi$

**while** *policy has not converged* **do**

    $o_{1:T} \sim D_{demo}$

    $s_0 \leftarrow 0$

    **for** $t = 1 : T$ **do**

        $a_{t-1} \sim \pi_\psi(a_{t-1} | s_{t-1})$

        $s_t \sim q_\phi(s_t | s_{t-1}, a_{t-1}, f_\phi(o_t))$

    Compute objective function $J$ from eq. (1)

    Update policy parameters $\psi \leftarrow \psi + \alpha \nabla_\psi J$

---

To make it more efficient, we use amortised inference. We directly define a policy $\pi_\psi(a_t | s_t)$ under the latent state of the world model. By composing the learnt world model and the policy, we can form a new generative model of the state sequence by the chain of $s_t \rightarrow a_t \rightarrow s_{t+1} \rightarrow a_{t+1} \ldots \rightarrow s_T$. Then we will get the same optimisation problem as eq. (8).

To sum up, in AIME, we use the same objective function – the ELBO – in both phases with the only difference being the source of the action sequence. We provide the pseudo-code for the algorithm in Algorithm 1 with the colour highlighting the different origins of the actions between the two phases.

## 4 EXPERIMENTS

To test our method, we need multiple environments sharing an embodiment while posing different tasks. Therefore, we consider the embodiment Walker of the DeepMind Control Suite (DMC Suite) Tunyasuvunakool et al. (2020), which poses three tasks: stand, walk and run. Following the common practise in the benchmark Hafner et al. (2020), we repeat every action two times when interacting with the environment. For the Walker environment, the true state includes both the position and the velocity of each joint and the centre of mass of the body. In order to study the influence of different observation modalities, we consider three settings for each environment: *MDP* uses the true state as the observation; *Visual* uses images as the observation; *LPOMDP* uses only the position part of the state as the observation, so that information-wise it is identical to the *Visual* setting but the information is densely represented in a low-dimensional form.

To generate the embodiment and demonstration datasets, we train a Dreamer Hafner et al. (2020) agent in the Visual setting for each of the tasks for 1M environment steps. Our Dreamer implementation can solve stand and walk quite well and efficiently. For the run task, the reward plateaus are around 600 but in the video, we can observe decent running behaviour. The exact performance of the demonstration dataset can be found at Table 1. We take the replay buffer of these trained agents as the embodiment datasets $D_{\text{body}}$, which contain 1000 trajectories, and consider the converged policy as the expert to collect another 1000 trajectories as the demonstration dataset $D_{\text{demo}}$. We only use 100 trajectories for the main experiments, and the left trajectories are used during ablation study. Besides the above embodiment datasets, we also study datasets generated by purely exploratory behaviour. To do this, we consider two settings. First, we use a random policy that samples uniformly from the action space to collect 1000 trajectories, and we call this a *random* dataset. Second, we train a Plan2Explore Sekar et al. (2020) agent for 1000 trajectories and label its replay buffer as the *p2e* dataset. Moreover, we also merge all the above datasets except the *run* dataset to form a *mix* dataset,

which is close to a practical setting when one has a lot of experience with one embodiment and use also the data they collect to train a model.

## 4.1 BASELINES AND IMPLEMENTATION DETAILS

We mainly compare our method with BCO Torabi et al. (2018a), specifically the BCO(0) variant. BCO(0) first trains an IDM from the embodiment dataset and then used the trained IDM to label the demonstration dataset and then uses Behavioural Cloning (BC) to recover the policy. We do not compare with other methods since they either require further environment interactions Torabi et al. (2018b); Li et al. (2022) or use a goal-conditional setting Pathak et al. (2018) which does not suit the locomotion tasks.

BCO(0) originally applies to the MDP setting. In order to adapt it to the POMDP setting, we use the common stacking trick Mnih et al. (2015). In particular, we stack 5 consecutive observations to form a state for BCO. For the Visual setting, before stacking, the observation first passes through a shared CNN encoder to extract features. The structure of the CNN is implemented as in Ha & Schmidhuber; Hafner et al. (2019). Both the IDM and the policy are implemented by an MLP with two hidden layers and 128 units for each layer. The activation functions are chosen to be ELU Clevert et al. (2015) for the hidden layers and Tanh for the output layer. Following the original paper, we split both datasets to $7:3$ for the training and validation dataset. We train the models with the Adam Kingma & Ba (2014) optimiser for 50 epochs and then use the model with the best validation loss. To ensure that we are not artificially limiting the performance by computational resources, we also tried increasing both the width and depth of the MLP, but did not find any increase in performance.

For our method, the RSSM implementation is largely following Dreamer-v1 Hafner et al. (2020) with continuous stochastic and deterministic variables. Although newer versions of Dreamer Hafner et al. (2021; 2023) offer some new tricks to improve performance, we choose not to use them for the sake of simplicity. We use a slightly larger state space for our experiment with 512 deterministic and 128 stochastic dimensions and find it generally eases the policy training process to collect the datasets. When decoding the image observation, we follow the same structure as Ha & Schmidhuber; Hafner et al. (2019), while for the low-dimensional observations, the decoder is implemented as Gaussian distribution with both mean and variance being parameterised. Except for the deterministic part of the state using a GRU cell Chung et al. (2014), all the other networks are MLPs with 2 hidden layers and 128 units of each layer. We do not use any free nats Hafner et al. (2020), KL scaling Hafner et al. (2020) and KL balancing Hafner et al. (2021) tricks in the literature to relax the constraint of the KL term. When decoding low-dimensional signals, we sometimes observed the decoder yielding a degenerate solution as found in Seitzer et al. (2022). We use their $\beta$-nll to remedy this problem, and since it re-weights the reconstruction term, we re-weight the KL term accordingly to maintain the balance. For phase 2 of AIME, we train the policy for 500 epochs and 100 gradient steps for each epoch. We report performance of the policy from the last epoch without any early stopping criteria.

To maximise efficient usage of computational resources, we directly use the trained world model of the Dreamer agent when conducting our experiments, except for the Random dataset where we train the model from scratch in phase 1. An important finding is that, although loading pre-trained weights generally helps performance, the method does not perform well when we also transfer the weights of the learnt policy. We conjecture that it is due to learnt policies being stuck in some local minima that they are unable to escape.

## 4.2 RESULTS

The main result of our comparison is shown in Figure 2. We can see that AIME largely outperforms BCO(0) in all the environment settings. AIME achieves the lowest performance on the Visual setting, but even that outperforms BCO(0)-MDP which can access the true states. We attribute the good performance to two reasons. First, the forward model has a better data utilisation rate than the inverse model because the forward model is trained to reconstruct whole observation sequences, while the inverse model only takes short clips of the sequence and only predicts the actions. Thus, the forward model has less chance to overfit and provides better generalisation. Second, by maximising the evidence, our method strives to find an action sequence that leads to the same outcome, not to recover the true actions. For many systems, the dynamics are not fully invertible. For example, if a human applies force to the wall, since the wall does not move, one cannot tell how much force is really

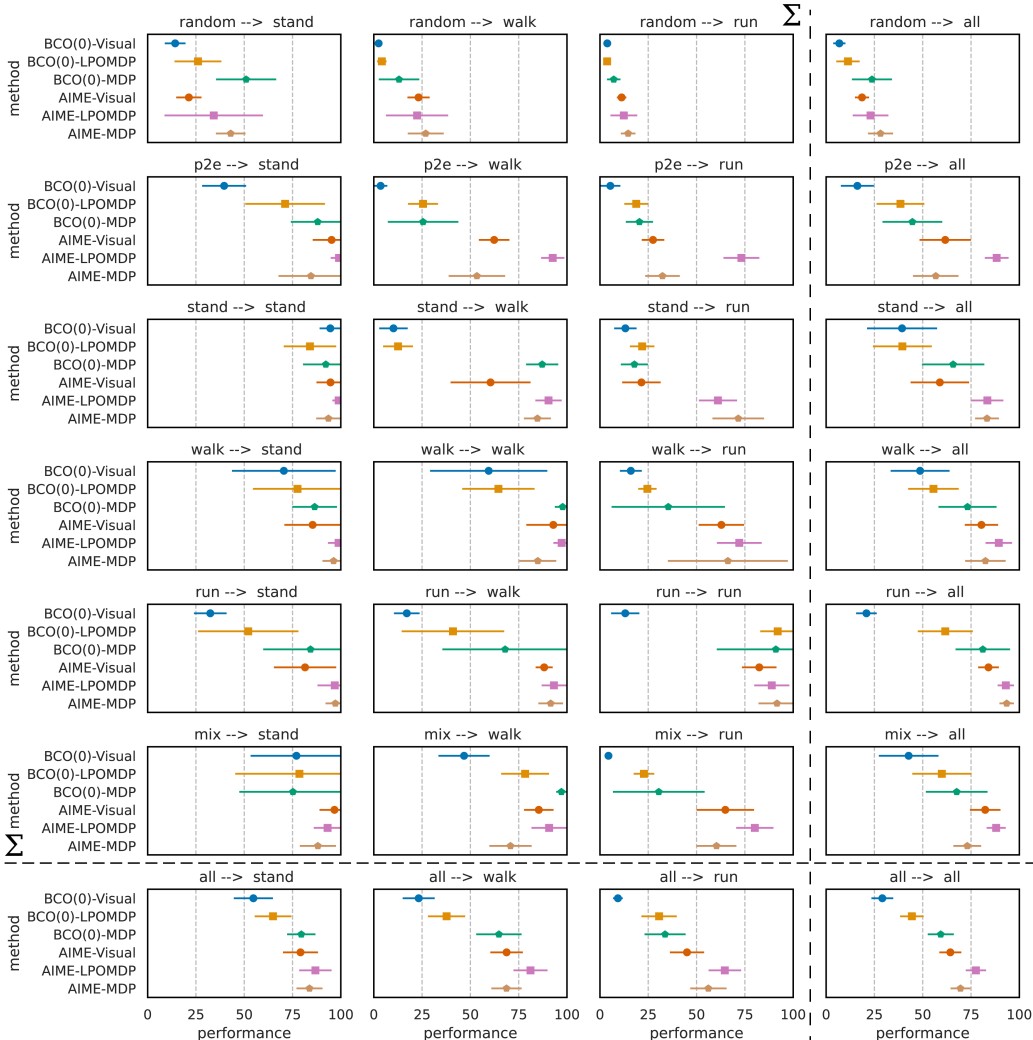

Figure 2: Performances on Walker. Each column indicates one task and its associated demonstration dataset, while each row indicates the embodiment datasets used to train the model. The title of each figure is named according to $D_{\text{body}} \to D_{\text{demo}}$. Numbers are computed by averaging among 10 trials and then normalized to the percentage of the expert's performance. The last row and column are averaged over the corresponding task or dataset.

applied by visual observation. The same situation applies to the Walker when certain joints are locked due to the singular pose. This same phenomenon is also discussed in Pathak et al. (2018).

**How does the choice of dataset influence the results?** First, as expected, for all the variants of methods, transferring within the same task is better than transferring between different tasks. In these settings, BCO(0)-MDP is comparable with AIME. However, AIME shines in cross-task transfer. Especially when transferring between run and walk tasks and transferring from stand to run, AIME outperforms the baselines by a large margin, which indicates the strong generalisability of a forward model over an inverse model. We also find that AIME makes substantially better use of exploratory data. AIME largely outperforms baselines when using the p2e dataset as the embodiment dataset and outperforms most variants when using the Random dataset as the embodiment dataset. Moreover, when transferring from the mix dataset, except for the MDP version, AIME outperforms other variants that train the world model on one of the mixed datasets individually. This showcases the scalability of a world model to be trained on a diverse set of experiences, which could be more valuable in real-world scenarios.

Figure 3: Ablation of number of demonstrations on transferring from p2e dataset. The performance is shown as the percentage of expert performance over 3 seeds.

**How is AIME influenced by the observation modality?** Compared with BCO(0), AIME is quite robust to the choice of observation modality, but it still makes a little difference. Although the observation provides the same information, we find AIME in the LPOMDP setting performs much better than in the Visual setting in all test cases. We attribute it to the fact that low-dimension signals have denser information and offer a smoother landscape in the evidence space than the pixels so that it can provide a more useful gradient to guide the action inference. Surprisingly, although having access to more information, AIME-MDP performs worse than AIME-LPOMDP on average. The biggest gap happens when transferring from p2e dataset. We conjecture this to the fact the RSSM world model of AIME-MDP is not trained well with the default hyper-parameters, but we defer further investigation to future work.

**How does the number of demonstrations influence the performance?** We conduct the ablation study on p2e embodiment dataset where we vary the number of demonstrations within $\{1, 2, 5, 10, 20, 50, 100, 200, 500, 1000\}$. The result is shown in Figure 3. BCO(0) struggles with low-data scenarios and typically needs at least 10 demonstrations to surpass its initial performance. In contrast, AIME demonstrates continual improvement with as few as 2 trajectories. This demonstrates the superior sample efficiency of our method. Moreover, the performance of AIME keeps increasing as more trajectories are provided beyond 100, which showcases the scalability of the method. We also notice that there is a dent in the performance curve at 5 demonstrations and we conjecture it is caused by bad demonstrations.

## 5 RELATED WORK

**Imitation learning from observations** Previous works on imitation learning from only observation can be roughly categorised into two groups, one based on IDMs Torabi et al. (2018a); Baker et al. (2022); Pathak et al. (2018) and one based on generative adversarial imitation learning (GAIL) Ho & Ermon (2016); Torabi et al. (2018b); Li et al. (2022). The core component of the first group is to learn an IDM that maps a state transition pair to the action that caused the transition. Torabi et al. (2018a); Baker et al. (2022) use the IDM to label the expert's observation sequences, then solve the imitation learning problem with standard BC. Pathak et al. (2018) extends the IDM to a goal-conditioned setting and communicates the task with keyframes of the goal trajectory. Different from these methods, our approach uses a forward model to capture the knowledge of the embodiment. In the second group of approaches, the core component is a discriminator that distinguishes the demonstrator's and the agent's observation trajectories. Then the discriminator serves as a reward function, and the agent's policy is trained by RL Ho & Ermon (2016). As a drawback, in order to train this discriminator the agent has to constantly interact with the environment to produce negative samples. Different from these methods, our method does not require further interactions with the environment, enabling zero-shot imitation from the demonstration dataset.

**Reusing learnt components in decision-making** Although transferring pre-trained models has become a dominant approach in natural language processing (NLP) Devlin et al. (2019); Radford et al. (2019); Bommasani et al. (2021) and has been getting more popular in computer vision (CV) He et al. (2022); Bommasani et al. (2021), reusing learnt components is less studied in the field of decision-making Agarwal et al. (2022). Most existing works focus on transferring policies Finn et al. (2017); Baker et al. (2022); Agarwal et al. (2022). On the other hand, the world model, a type of powerful perception model, that is purely trained by self-supervised learning lies behind the recent progress of model-based reinforcement learning Ha & Schmidhuber; Hafner et al. (2019; 2020; 2021; 2023); Łukasz Kaiser et al. (2020); Schrittwieser et al. (2020). However, the transferability of these

world models is not well-studied. Sekar et al. (2020) learns a policy by using a pre-trained world model from exploration data and demonstrates superior zero-shot and few-shot performance. We improve upon this direction by studying a different setting, i.e. imitation learning. In particular, we communicate the task to the model by observing the expert while Sekar et al. (2020) communicates the task by a ground truth reward function which is less accessible in a real-world setting.

## 6 DISCUSSION & CONCLUSION

In this paper, we present AIME, a model-based method for imitation from observations. The core of the method exploits the power of a pre-trained world model and inverses it w.r.t. action input by taking the gradients. On the Walker embodiment from the DMC Suite, we demonstrate superior performance compared to baselines, even when some baselines can access the true state.

Although AIME performs well, there are still many limitations. First, humans mostly observe others with vision. Although AIME works quite well in the *Visual* setting, there is still a large gap compared with the LPOMDP setting where the low-level signals are observed. We attribute this to the fact that the loss surface of the pixel reconstruction loss may not be smooth enough to allow the gradient method to find an equally good solution. Second, in this paper, we only study the most simple setting where both the embodiment and sensor layout are fixed across tasks. On the other hand, humans observe others in a third-person perspective and can also imitate animals whose body is not even similar to humans'. Relaxing these assumptions will open up possibilities to transfer across different embodiments and even directly from human videos. Third, for some tasks, even humans cannot achieve zero-shot imitation by only watching others. This may due to the task's complexity or completely unfamiliar skills. So, even with proper instruction, humans still need to practise in the environment to solve some tasks. This motivates an online learning phase 3 as an extension to our framework. We defer these topics to future work.

### ACKNOWLEDGMENTS

We would like to acknowledge Elie Aljalbout for the insightful discussion during the initial stage of the project and Botond Cseke for mathematical support.

### CONNECTION TO THE WORKSHOP

In this paper, we propose AIME, an algorithm that uses a trained world model, a prior computation, to imitate expert behaviours from only observations. We hope this paper demonstrates the great potential of transferring a learnt world model, incentivises more people to work in this direction and encourages researchers to also share their learnt world model to contribute to the community.

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

## A    DATASET INFORMATION

Here we provide extra information about the datasets. The expert return which we normalised against is shown in Table 1.

Table 1: Average expert return of each demonstration dataset

| $D_{\text{demo}}$ | Average return |
| --- | --- |
| stand | 957.87 |
| walk | 943.79 |
| run | 604.10 |

## B    SAMPLES OF TRAINING CURVES

In this section, we present some representative training curves of AIME's phase 2 from our experiments in Figure 4. The first three figures show the transfer from the mix dataset to the run task

in the three settings which are the typical success cases of AIME. During the course of training, ELBO is maximised towards convergence and the MSE between the generated actions and the true actions decreases. We can also see that for the MDP and LPOMDP settings, the converged ELBO is lower than the ELBO when evaluated with the true action sequence, indicating there is still space for improvement. However, for the Visual setting, the converged ELBO exceeds the one with true actions, which should be attributed to the over-fitting of the world model from phase 1. The last three figures show the transfer from the random dataset to the three tasks in the Visual settings which we consider as failure cases. For the stand and walk tasks, none of the metrics are converging. For the run task, we can observe a severe over-fitting starting from the beginning of the training, and the MSE keeps increasing. We conjecture these are all due to the less well-trained world models.

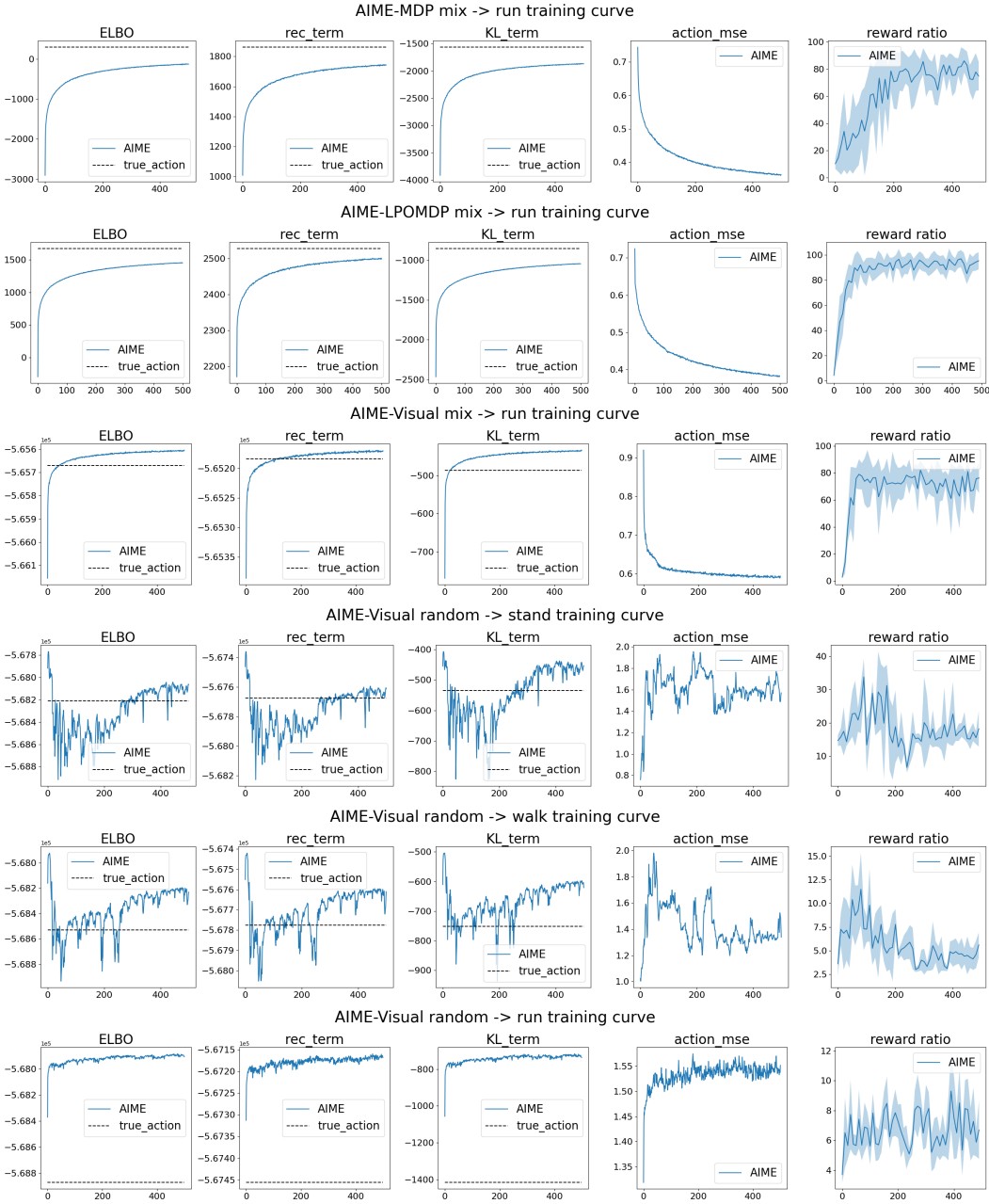

Figure 4: Samples of training curve in phase 2 of AIME. The first three showcase the typical successful training curves, while the remaining three demonstrate the failure cases. The true_action is referring to evaluating the trajectories with the true action sequence.

