# OpenReview forum: "Action Inference by Maximising Evidence: Zero-Shot Imitation from Observation with World Models"
_ICLR.cc/2023/Workshop/RRL — RRL 2023 Poster_

### Official Review · Reviewer_cnUa · 2023-02-26
**Interesting work, but unclear contribution**

**Rating:** 2
**Confidence:** 2

**Review:**

# Summary

This paper presents AIME, which is an imitation learning method that learns a world model from past experience, then uses that model to infer actions from an observation-only demonstration.

---

## Comments / Questions
-  AIME can be used to leverage a world model which is trained on data from multiple tasks
- It is not clear to me what benefit AIME has over BCO and model-based reinforcement learning approaches.  I think the authors could take some time in Section 3 to lay out the differences explicitly.
- The task used for evaluation is relatively simple, so it would be nice to see how AIME compares to other algorithms in more complex settings where it is extremely difficult to learn a world model
- Could mathematical notation in Section 3 be condensed?

---

### Official Review · Reviewer_Cnqa · 2023-03-01
**"Transfer learning" in agents by imitating experts**

**Rating:** 4
**Confidence:** 5

**Review:**

This paper proposes the AIME framework - Action Inference by Maximizing Evidence - a 2 step approach to train reinforcement learning agents where the agent first learns a world model from actions and observations, and then imitates the behavior of an expert based on a set of observations.

The theoretical and implementation aspects of the methodology are crisply motivated and explained. The proposed approach is tested on multiple representative environments. The choice of baselines and experimental setup is meticulously documented. The results clearly showcase the generalizability and efficacy of the approach. The authors discuss interesting observations and outline potential future directions.

The paper is concise and written in a simple and understandable manner. The authors also compare their approach against related work in the fields of imitation learning and reusing pretrained components and succinctly outline the unique contributions of the paper.

The proposal is interesting and evokes comparisons to transfer learning techniques that have proven incredibly effective in sequence modeling research, where a pretrained model learns general rules about the domain and the knowledge can be transferred in a few-shot manner to downstream tasks. The authors discuss these similarities and present the gaps in current RL research on corresponding ideas in the related works section. I’d encourage the authors to continue exploring these themes!